



# Measuring FeO variation using astronomical spectroscopic observations

Stefanie Unterguggenberger[1], Stefan Noll[1], Wuhu Feng[2,3], John M. C. Plane[2], Wolfgang Kausch[4,1], Stefan Kimeswenger[5,1], Amy Jones[6,1], and Sabine Moehler[7]

[1]Institut für Astro- und Teilchenpyhsik, Universität Innsbruck, Technikerstr. 25/8, 6020 Innsbruck, Austria
[2]School of Chemistry, University of Leeds, Woodhouse Lane, Leed LS 9JT, United Kingdom
[3]National Centre for Atmospheric Science, University of Leeds, Woodhouse Lane, Leed LS 9JT, United Kingdom
[4]Institute for Astrophysics, University of Vienna, Türkenschanzstr. 17, 1180 Vienna, Austria
[5]Universidad Católica del Norte, Avenida Angamos 0610, Antofagasta, Chile
[6]Max Planck Institute for Astrophysics, Karl-Schwarzschild-Str. 1, 85748 Garching, Germany
[7]European Southern Observatory, Karl-Schwarzschild-Str. 2, 85748 Garching, Germany

Correspondence to: Stefanie Unterguggenberger (stefanie.unterguggenberger@uibk.ac.at)



## Abstract

Airglow emission lines of OH, $O_2$, O and Na are commonly used to probe the MLT (mesosphere/lower thermosphere) region of the atmosphere. Furthermore, molecules like electronically excited NO, NiO and FeO emit a (pseudo-) continuum. These continua are harder to investigate than atomic emission lines. So far, limb-sounding from space and a small number of ground-based low-to-medium resolution spectra have been used to measure FeO emission in the MLT. In this study the medium-to-high resolution echelle spectrograph X-shooter at the Very Large Telescope (VLT) in the Chilean Atacama desert ($24°\ 37′$ S, $70°\ 24′$ W, $2\,635$ m) is used to study the FeO pseudo-continuum in the range from 0.5 to $0.72\,\mu m$ based on $3\,662$ spectra. Variations of the FeO spectrum itself, as well as the diurnal and seasonal behaviour of the FeO and Na emission intensities are reported. These airglow emissions are linked by their common origin, meteoric ablation, and share $O_3$ as a common reactant. Major differences are found in the main emission peak of the FeO airglow spectrum between 0.58 and 0.61 µm, compared with a theoretical spectrum. The FeO and Na airglow intensities exhibit a similar nocturnal variation, and a semi-annual seasonal variation with equinoctial maxima. This is satisfactorily reproduced by a whole atmosphere chemistry climate model, if the quantum yields for the reactions of Fe and Na with $O_3$ are $13\pm3\%$ and $11\pm2\%$, respectively. However, a comparison between the modelled $O_3$ in the upper mesosphere and measurements of $O_3$ made with the SABER satellite instrument suggests that these quantum yields may be a factor of $\sim2$ smaller.

## 1   Introduction

The mesosphere lower thermosphere (MLT) is a layer of the atmosphere located between $\sim70$ to 110 km above the surface (Plane et al., 2015). The ablation of cosmic dust particles entering the atmosphere is the main source of the metals (Fe, Si, Mg, Na, K and Ca) in the MLT (e.g. Plane, 1991; Plane et al., 2015). These metals have been measured by space-based observation (e.g. Hedin & Gumbel, 2014; Fan et al., 2007; Evans et al., 2010;



Dawkins et al., 2015; Langowski et al., 2015) and ground-based lidar (e.g. Höffner & Friedman , 2004, 2005; Gardner et al., 2005; Gardner et al., 2011; Lübken et al., 2011; Yi et al., 2009; Friedman et al., 2013) over a range of latitudes.

There are currently around 20 lidar stations worldwide observing Na, about 10 stations studying Fe, and three each for K and $Ca/Ca^+$ according to Plane et al. (2015). Yi et al. (2009) and Gardner et al. (2005) conducted studies of the variations of the Na and Fe layers at Wuhan, China ($30° 34'$ N, $114° 17'$ E) and the South Pole, respectively. Yi et al. (2009) observed semi-annual variations of both Na and Fe, while Gardner et al. (2005) showed that at the South Pole there was a significant annual variation in both metals, with a summer minimum in Na and Fe density.

Another method that has been used to study the meteoric metals uses low-to-medium resolution spectroscopy of the nightglow. Na is a good candidate for this technique since its chemiluminescent emission can be measured as a doublet at 0.5890 and 0.5896 µm. Atomic Na reacts with $O_3$ and forms NaO. The latter then reacts with atomic O to yield Na in the excited $Na(^2P_{3/2,1/2})$ states. In fact, both the ground state of $NaO(^2\Pi)$ and the first excited state $NaO(^2\Sigma^+)$ are involved, but overall the quantum yield for photon emission arising from the Na + $O_3$ reaction should be 16.6% on the statistical basis of correlating electronic states, which is in accordance with rocket-borne and laboratory measurements (Plane et al., 2012).

Chemiluminescent emission from electronically excited FeO appears as pseudo-continuum at wavelengths between 0.5 and 0.72 µm, termed the "orange arc" bands. The emission is produced by the reaction Fe + $O_3$ with a quantum yield of ∼2%, measured in a laboratory study (West & Broida, 1975). Being part of the night-sky continuum emission implies that it is also contaminated by other airglow continua (e.g. NO+O (Khomich et al., 2008) and NiO (Evans et al., 2011)), and astronomical sources such as scattered star- and moonlight, and zodiacal light.

Two previous studies have employed spectroscopy to measure the FeO airglow emission. Evans et al. (2010) used the OSIRIS (Optical Spectrograph and InfraRed Imager System) spectrometer on the Odin satellite to detect the feature in the night-sky continuum. They





found that the spectral shape of FeO from their study matched the laboratory spectrum of West & Broida (1975). Gattinger et al. (2011) calculated a theoretical spectrum, using laboratory data and the spectra obtained from OSIRIS. These studies were followed by Saran et al. (2011) who used the ground-based ESI spectrograph (resolving power $\lambda/\Delta\lambda$ $\sim$7 000, $\lambda$ = 0.39 to 1.1 µm) at the Keck observatory (19° 49.6′ N, 155° 28.5′ W) to detect FeO contribution to the night-sky spectrum. Data from five nights in March 2000 and four nights during October of the same year were presented. This study showed that the diurnal behaviour of Na and FeO are closely related. This is not surprising since the two emissions share the same source, meteoric ablation of cosmic dust and involve reactions with $O_3$. Saran et al. (2011) investigated the intra-annual change of the FeO emission strength with spectra taken in March and October 2009 as well as spectra from February and June 2010 taken at Kitt Peak (31° 57.5' N, 111° 35.8' W). They found a clearly visible change in the FeO pseudo-continuum with the seasons.

In this paper we investigate the FeO airglow continuum with respect to the already well studied Na emission. We used all publicly available data from the X-shooter echelle spectrograph at Cerro Paranal (24° 37′ S, 70° 24′ W, 2 635 m) (Vernet et al., 2011) from October 2009 to March 2013, compiling a sample of 3 662 spectra with higher resolution than previous studies (Evans et al., 2010; Saran et al., 2011). With our sample it is possible to investigate differences between the observed spectra and the theoretical one provided by Gattinger et al. (2011) (see Sect. 3.1.2). The resolution of X-shooter enables us not just to distinguish between the different features of the FeO emission but also to study finer structures within the FeO emission. The diurnal variability (3.2) and seasonal variability (3.3) of the FeO and Na emissions are then reported, and the seasonal variability is compared with the predictions of a whole atmosphere chemistry climate model (3.4).



## 2 Observations and data set

The spectra were taken with the Very Large Telescope (VLT) operated by the European Southern Observatory (ESO). These four 8-meter class telescopes are located in the Chilean Atacama desert at an altitude of 2 635 m on the top of Cerro Paranal. Since the VLT is an as-
5 tronomical research facility we do not have dedicated airglow observations, but use the observed astronomical data. The Earth's atmosphere leaves its fingerprint in every spectrum taken at the facility. Especially valuable are long-slit spectra since they contain a portion of the night-sky in addition to the astronomical science object. X-shooter is a medium-to-high resolution spectrograph with a field of view between $0.4 \times 11$ arcsec$^2$ and $5 \times 11$ arcsec$^2$. It is
10 a unique instrument since it covers a wavelength range from 0.3 to 2.5 μm simultaneously with a resolving power ranging from 3 000 to 18 000, depending on the covered wavelength range and slit width. While the wavelength range is covered with one shot, the spectrum is split into three so-called arms: the UV to blue part of the spectrum (UVB, 0.3-0.56 μm), the "visual" regime (VIS, 0.56-1.02 μm), and the near infrared (NIR, 1.02-2.5 μm). To study FeO
emission (0.5 - 0.72 μm) the VIS arm was used. X-shooter is a frequently used instrument, so there is comprehensive coverage over several years. Figure 1 shows the available data that had a minimum exposure time of 10 minutes, and passed additional quality checks after the data reduction. An exposure time of 10 minutes provides a sufficient signal-to-noise (S/N) to reliably detect FeO emission. For the data reduction, an adapted version of pipeline
v2.6.8 of the ESO public pipeline (Modigliani et al., 2010) was used, which allowed us to use data of almost all observing modes to determine the sky spectrum (Vernet et al., 2011; Noll et al., 2015). The reduction results in a two dimensional (2D) sky spectrum which was then collapsed to a 1D sky spectrum using a median along the spatial direction. The absolute intensity calibration was performed using spectra of spectrophotometric standard stars
(Moehler et al., 2014), which were corrected for molecular absorptions (Smette et al., 2015; Kausch et al., 2015) and atmospheric extinction (Patat et al., 2011). For more details see Noll et al. (2015). In addition to Noll et al. (2015) we also performed a continuum correction. We corrected for the influence of residuals related to the echelle orders introduced by the



pipeline.

Our quality checks leave us with a sample of 3 662 spectra (see Fig. 1). Over the whole period from October 2009 and March 2013, 33%, 32% and 22% of the data were taken in 2011, 2010 and 2012, respectively, while 2009 and 2013 account for 13%. Each month from April to August contributes between 5 to 8% to the sample. The most prominent months are December and January with 10%, and as well as March and September with 13%. This uneven distribution is due to the observation schedule of the VLT.

Four months of UVB arm spectra (January, April, July and October 2010) were used as a control sample in Sect. 3.1.2. This sample consists of 223 spectra with each of them having a counterpart in the VIS arm.

## 3 Methods and Results

Compared to other airglow emissions such as Na and O, FeO is only a faint pseudo-continuum component. Consequently, there are two issues to be taken care of when measuring the FeO emission intensity in the X-shooter spectra: first, sufficient integration time (see Sect. 2); second, the subtraction of any other emission in the wavelength region of the FeO emission. This includes contributions from zodiacal light, scattered star- and moonlight. These components were corrected by using the sky model described in Noll et al. (2012) and Jones et al. (2013). The airglow emission lines were corrected by fitting a Gaussian function to define the width of each line. Median values of predefined continuum windows, with close proximity to each emission line, were used to interpolate the corresponding wavelength range. Figure 2 illustrates the result before and after the subtraction of the airglow emission lines. The structure of the FeO spectrum can be clearly identified in Fig. 2, panel (b). The spectrum has an exposure time of roughly 1 hour and is in the medium range ($\lambda/\Delta\lambda \approx 7\,450$) of the X-shooter resolving power.

Figure 2 also illustrates that the most prominent section of the FeO emission is between 0.58 and 0.61 μm, where the main emission peak is located. In this window only Na and OH(8-2) have to be subtracted. The other parts of the FeO emission bands have a higher





contamination of airglow emission lines and have a lower S/N. Hence, the integrated intensity measurements presented below are from this interval around the main peak. The continuum for the intensity measurements is defined from the minima on each side of the main peak. Since the minima are narrow regions and their depth is variable, they are the dominant error source. The magnitude of this error was estimated using a Monte Carlo method. By varying the width and position of the continua on both sides of the peak and calculating the respective intensities, the uncertainty in the integrated peak intensity was derived. The winter months, June to August, contain fewer spectra than the rest of the year (see Fig. 1). Hence we tested if any bias was introduced by the uneven distribution. We tested our results with respect to the data distribution over the year by introducing equally spaced bins. Using these bins, our results only changed within the error (see Sect. 3.3). In comparison, Saran et al. (2011) investigated the FeO intensity in a window between 0.56 and 0.62 μm. They subtracted the Na and OH(8-2) emission and the continuum level, which is according to Gattinger et al. (2011) defined at 0.5 μm from their selected wavelength range. We estimated an error for the FeO main peak measurements of 6% by testing the possible influence of other continuum components (NiO, NO+O) as well as the choice of the side minima of the main peak.

The intensity of the Na doublet was measured by summing the intensities of Na $D_1$ and Na $D_2$ and subtracting the underlying continuum. The continuum was obtained individually for each line by a fit of a Gaussian to the lines and using the offset of the obtained parameters as continuum value. FeO and Na intensity measurement were corrected for the van Rhijn effect (see Noll et al., 2015).

In the following sections we investigate the deviation of the observed FeO spectrum from the one derived by Gattinger et al. (2011) with respect to the FeO emission strength intensity (Sect. 3.1). In Sect. 3.2 the diurnal variability of the FeO emission in the four different seasons is examined and compared with the results from Na. In Sect. 3.3 the seasonal behaviour of the FeO and Na intensities are discussed and compared with the study of



Takahashi et al. (1995). Finally, we compare in Sect. 3.4 the observational results to the predictions of a Whole Atmosphere Chemistry Climate Model (WACCM).

## 3.1 Variability of the FeO spectrum

### 3.1.1 Methods

To investigate the variability of the FeO emission, we sorted the 3 662 available spectra according to their main peak intensity by dividing them into five bins (quintiles) and merge these spectra into a median spectrum for each bin. Figure 3a illustrates the different median FeO spectra. The first quintile spectrum (blue line in Fig. 3), is the one with the lowest intensity, whereas the fifth quintile spectrum (red) is the one with the median spectrum of the highest intensity. The fifth quintile spectrum is well represented in all years, with the most prominent months March, April, September and October. The spectra of the first and second quintile bin are evenly distributed over all the years and mainly originate from the months December and January (50% of the total number of spectra).

To compare the median spectra to the theoretical spectrum of Gattinger et al. (2011) we normalize the theoretical spectrum to the mean intensity in the main peak. Then the median spectra were normalized by scaling their main peak to the unit area of the theoretical main peak (see Fig. 3b). In the following we concentrate on the first and fifth quintile spectrum. For Fig. 3c we subtracted the theoretical spectrum from the median spectra.

The regions shaded in grey in Fig. 3 are regions were deviations between the theoretical and the median spectra have to be taken with caution since residuals of strong absorption features from molecular absorption ($O_2$), scattered star-, moon- and zodiacal light or airglow emission lines may not have been properly corrected.

### 3.1.2 Results

In general, the five quintile spectra show a good overall agreement in terms of shape and emission pattern (see Fig. 3b). We found that they can be easily reproduced by any other quintile spectrum by multiplying them with a constant factor (see Fig. 3b). This indicates





either a continuum contribution of e.g. NO+O and NiO that scales directly with the FeO emission or that the continuum contribution is very weak and hence does hardly contribute to the FeO spectrum. It should also be kept in mind that by subtracting the different components of the sky model and the residuals of the X-shooter pipeline an overcorrection is another possibility.

Comparing our median spectra with the theoretical one derived from Gattinger et al. (2011) we find a good agreement in general (Fig. 3b). However, there are significant differences within and in close proximity to the main peak (0.585 and 0.615 µm) as illustrated in Fig. 3c. The theoretical reference spectrum shows a decrease of the main peak from 0.595 to 0.585 µm whereas we find the main peak to be extended until 0.585 µm. Part of the excess could be attributed to the residuals of Na emission lines and the OH(8-2) band, but the excess window is broader than these features. Furthermore, the theoretical spectrum shows structures within the main peak (0.595 and 0.600 µm). Our median spectra show less structure. The higher the intensity of the quintile spectrum the smoother the main peak, which is indicated by the excess in flux at 0.59 µm in Fig. 3c. One more deviation can be seen between 0.615 and 0.626 µm. A possible explanation for the deviation could be a contamination by NiO. Evans et al. (2011) showed that NiO emission can be found in the same wavelength regime as the FeO emission. They found a variation in the ratio of NiO/FeO emission from 0.05 to 0.3 using data from the OSIRIS spectrograph, and a ratio of 2.3 was obtained on board a space shuttle (see Gattinger et al., 2011). The NiO emission overlaps with FeO emission in the VIS arm and cannot be separated. We tested our sample for NiO emission by using spectra from the UVB arm. The FeO emission starts at 0.5 µm (Gattinger et al., 2011), whereas NiO extends further into the blue down to about 0.465 µm (Burgard et al., 2006). Hence we studied the UVB for distinctive features and compared them to the NiO spectrum by Burgard et al. (2006). We found one peak between 0.495 and 0.508 µm. Using this peak to scale the laboratory NiO spectrum accordingly we find a maximum contribution of 31% to the main peak, while the average contribution is in the order of ∼2%. However, this estimate has to be treated with caution, since we could not find any correlation between the possible NiO emission and the FeO emission or any other distinct NiO



emission signatures in either the UVB or VIS arms.

Scaling the main peak emission from Gattinger et al. (2011) to the whole spectrum we obtain a value of 3.9%. To compare this to our measured spectra we reconstructed a FeO spectrum from our sample. First, we calculated a median spectrum of the whole sample. The VIS arm spectra do not extend further to the blue than 0.56 µm, so we extended it from 0.56 µm to 0.5 µm by scaling the corresponding part of the theoretical spectrum accordingly. The grey-shaded regions in Fig. 3 were replaced in the same way. Our median spectrum from 0.68 to 0.72 µm is heavily influenced by further airglow emission. Hence this part was also replaced by the scaled theoretical spectrum. This procedure provided us with a FeO spectrum, where the main peak amounts to 3.3±0.8%. For the estimated error we took scaling errors of the theoretical spectrum to the overall median spectrum into account as well as contamination by the NiO and NO+O continuum.

The same tests were done with a FeO laboratory spectrum by Jenniskens et al. (2000). This spectrum showed a shift in the main peak of 5 nm compared to our median spectra. Furthermore, the main peak in Jenniskens et al. (2000) is more prominent than in our median spectra. The ratio of the main peak to the whole spectral range would be 12% which is a factor of four larger than what we estimated from our median spectra. Also the side peaks on the red side of the main peak are hardly traced at all. The overall agreement of the laboratory spectrum to our median spectrum is worse than the agreement with the theoretically derived one.

## 3.2  Diurnal variability

### 3.2.1  Methods

The sample is sufficiently well distributed over the different seasons to study the nocturnal behaviour of the FeO and Na emission. For clearer statistics the night was divided into five bins (in local time) by using 21:00, 23:00, 1:00 and 3:00 as delimiters. The number of data points within each bin is highly variable and varying between 110 and 316. In general, the bins at the beginning and end of the night are the low statistic bins. The error bars were



derived from the standard deviations of the measurements. For a clearer comparison, the relative intensities are plotted normalised to the nightly mean. All seasons we refer to are related to the southern hemisphere.

### 3.2.2 Results

In Fig. 4 we illustrate the nocturnal behaviour of FeO and Na during the four seasons at Cerro Paranal in the Atacama desert. In general, the Na and FeO emission show similar diurnal variation within their combined errors, i.e. Figs. 4a, c and d. The intensities of FeO and Na decline at the beginning of the night and rise towards sunrise. These panels also show that the relative intensity of Na is strongest at the end of the night, while the relative FeO intensity is either at the same level or lower than at the beginning of the night. Figure 4b shows a steady decline of the relative FeO and Na intensities throughout the night. The decline in relative intensity is found in the same period of the year in 2011 and 2012, while 2010 shows a rise towards the morning. The data from September to November in 2011 do not comply with 2010 and 2012, since there is a steep decline towards the final bin of the night.

### 3.3 Seasonal variability

### 3.3.1 Methods

The annual and semi-annual variations of the FeO and Na intensities were determined by fitting the data to the following expression (e.g. Takahashi et al., 1995):

$$F(t) = A_0 + A_1 \cdot \cos\left(\frac{2\pi(t - \varphi_1)}{365}\right) + A_2 \cdot \cos\left(\frac{2\pi(t - \varphi_2)}{182.5}\right) \qquad (1)$$

where $A_1$ and $A_2$ are the amplitudes of the annual and semi-annual oscillations, respectively, and $\varphi_1$ and $\varphi_2$ describe their corresponding phases. The parameter $A_0$ denotes the annual mean. The best-fit approach ($\chi^2_{\min}$) relies on the grid size and does not provide

Discussion Paper | Discussion Paper | Discussion Paper | Discussion Paper |





uncertainties. Hence, we used Bayesian statistics to confirm the regression best fit and to derive the uncertainties for our results.

For this, we created a grid containing different combinations of the five parameters $A_0$, $A_1$, $A_2$, $\varphi_1$ and $\varphi_2$. $\varphi$ was varied from 0 to 365 days and the amplitudes A between 0 and 0.65 × the maximum intensity of the FeO or Na emission. For each of the created models above, we calculated the likelihood as follows:

$$L = e^{-\left(\chi^2 - \chi^2_{\min}\right)/2} \tag{2}$$

where $\chi^2$ is the value and $\chi^2_{\min}$ is the minimum $\chi^2$ within the grid of parmeter sets. The results of Eq. 2 are then used to obtain the probability of the best parameter set by,

$$P_{\mathrm{par}} = \frac{\Sigma L_{\mathrm{par}}}{\Sigma L}. \tag{3}$$

$L_{\mathrm{par}}$ are the likelihoods, where one parameter of Eq. 1 is held constant and all others are varied. $P_{\mathrm{par}}$ is the derived probability of a given parameter set. Equation 3 does not contain the prior term since all models are assumed to have the same weight which results in a flat prior. Using Eq. 3 we are able to find a maximum within the parameter space corresponding to the most likely parameter set.

Table 1 shows the results of these calculations, with the mean value being a weighted mean and $\sigma$ giving the standard deviation of the probability distribution.

### 3.3.2 Results

Figure 5 shows the measured intensities for the FeO main peak (Fig. 5a) and Na (Fig. 5b). The solid lines show the results of the least squares fit of Eq. 1, and the square the Bayesian solution.

FeO has a stronger semi-annual amplitude in the best-fit solution, whereas the amplitudes for Na are equal and only show a dominating semi-annual amplitude using the Bayesian



analysis. The relative intensity change derived between $A_1$ and $A_2$ in Na ($A_1$: 25%, $A_2$: 30%) tends to be less prominent than the change in FeO ($A_1$: 17% $A_2$: 27%). Taking the errors of the Bayesian analysis into account (see Tbl. 1) we cannot exclude that $A_1$ and $A_2$ are equally strong for FeO and Na.

Comparing the phases of Na and FeO derived with both methods we find them to be in good agreement within the error. FeO and Na show their global minimum at the beginning of February and their second minimum at the end of August, while the maxima can be found at the end of May and the beginning of November when combining the annual and semi-annual components.

The results for the Na emission can also be compared with the study of Takahashi et al. (1995), which used the same model (see Eq. 1) and was carried out at a similar latitude (Cachoeira Paulista; 22° 39′ S). They reported Na amplitudes relative to the annual mean of $A_1$ = 28% and $A_2$ = 41%. The semi-annual amplitude is clearly the dominant component. Fukuyama (1976, 1977) and Wiens & Weill (1973) examined the latitude dependency of the

relative strengths of the annual and semi-annual amplitude of the emission intensities of Na and other MLT airglow emissions. Their studies showed that the semi-annual oscillation decreases at higher latitudes. Comparing Na relative amplitudes of Cerro Paranal ($A_1$: 25%, $A_2$: 30%) with Cachoeira Paulista ($A_1$: 28%, $A_2$: 41%), we find that the results for $A_1$ are in good agreement. On the other hand, the difference in latitude is probably too small to

explain the discrepancies in $A_2$. However, differences in the sampling, observing period or longitude could be responsible for equally strong $A_1$ and $A_2$ at Cerro Paranal. Comparing the phases of sodium from Takahashi et al. (1995) with our study, the annual and the semi-annual phase are in agreement within the errors.

## 3.4  Models

### 3.4.1  Methods

Here we use the Whole Atmosphere Community Climate model (WACCM), in which the injection through meteoric ablation and chemistry of Na (Marsh et al., 2013) and Fe (Feng





et al., 2013) have been included, to simulate the Na and FeO emission intensities at Cerro Paranal in the Chilean Atacama desert. WACCM is a coupled chemistry climate model which consists of a fully interactive chemistry and dynamics from the surface to an upper boundary at $6.0 \times 10^{-6}$ hPa ($\sim$140 km) (Marsh et al., 2013). The model was run with specified dynamics (termed as SD-WACCM) using ECMWF-Interim analyses (Dee et al., 2011) below 50-60 km from 2004-2015. The horizontal resolution is $1.9°$ latitude $\times$ $2.5°$ longitude, with 88 vertical levels with a vertical resolution of approximately 3.5 km in the MLT region. The photolysis, neutral and ion-molecule chemistry as well as meteoric input function are described in Plane et al. (2015).

The airglow intensities were calculated by integrating the FeO and Na volume emission rates between 80 and 100 km. The volume emission rates were using the modelled concentrations of Fe, Na and $O_3$, and the reaction rate coefficients for Fe + $O_3$ and Na + $O_3$ at the modelled temperatures. The airglow intensities were calculated daily at local midnight from 1 January 2009 to 31 December 2015, thus covering the period of the observations.

### 3.4.2 Results

Figure 6 shows a comparison of the nightly-averaged observations of FeO emission, considering a 3.9% contribution of the main peak to the total emission (see Sect. 3.1.2), with the model at local midnight. The modelled intensities are scaled by a factor of 0.13 to produce a correlation plot (lower panel) with a slope of 1 when forced through the origin. That is a quantum yield of $13\pm3$% for the Fe + $O_3$ reaction, which is much closer to the laboratory measurement of West & Broida (1975) (2-6%) than the previous modelling study of Saran et al. (2011) (100%). The error of the quantum yield was estimated by convolving the 22% uncertainty in the Fe + $O_3$ reaction rate coefficient (Helmer et al., 1994), a 6% uncertainty in the absolute calibration of the observed FeO* intensity, and a 2% uncertainty from the correlation plot in Fig. 6. Figure 7 is the analogous plot for the Na airglow emission intensity. The modelled intensities are scaled by a factor of 0.11 to produce a correlation plot (lower panel) with a slope of 1 when forced through the origin. The corresponding quantum yield

Discussion Paper | Discussion Paper | Discussion Paper | Discussion Paper |





of $11\pm2\%$ for the Na + $O_3$ reaction is really an effective quantum yield, since photons at 0.5890 and 0.5896 µm are produced from both the $^2\Pi$ and $^2\Sigma^+$ states of NaO reacting with O (Plane et al., 2012). The uncertainty in the Na quantum yield is estimated by convolving the 12% uncertainty in the Na + O3 reaction rate coefficient (Plane et al., 1993) with a 6% absolute calibration uncertainty and 2% uncertainty in the correlation plot (see Fig. 7). In both cases, the model captures well the semi-annual seasonal variation of the airglow intensities, with equinoctial maxima which correspond to peaks in MLT $O_3$ density (Thomas et al., 1984). Note that the autumnal peak of the FeO emission intensity in March/April is larger than the vernal peak in September/October. The reason for this is elucidated in Fig, 8, which shows that the injection flux of meteoric Fe peaks in March/April (Feng et al., 2013). This coincides with maxima in both the atomic Fe and $O_3$ density in WACCM (shown at a representative height of 87 km in Fig. 8). Since the FeO emission intensity is proportional to the product of [Fe] and [$O_3$], it maximises in autumn between March and May. In September to November, the secondary $O_3$ peak is offset by a low Fe density, as the Fe injection flux is near minimum. The same considerations apply to the seasonal variation of the Na emission intensity. A final point to note is that there is evidence that WACCM may underestimate the $O_3$ concentration in the airglow region between 86 and 89 km, as a result of insufficiently rapid downward transport of atomic O from the thermosphere (Smith et al., 2015). We have therefore compared the WACCM night-time $O_3$ at 24° S with SABER for the period of the present study. The modelled $O_3$ is on average between 45 and 56% lower than SABER. This suggests that the quantum yields estimated in this study could be lower by a factor of $\sim2$. This would bring the FeO quantum yield into even better agreement with the laboratory study of West & Broida (1975). The Na quantum yield would then be close to lower end of the range of 0.05 - 0.2 that was estimated from a combined rocket-borne and lidar experiment (Clemesha et al., 1995).





## 4 Conclusions

Astronomical data are a valuable addition to the study of airglow emission. Astronomical facilities provide long-time archives of medium-to-high resolution spectra. Using the X-shooter archive data we compiled a sample of 3 662 spectra showing FeO emission with a minimum exposure time of 10 min.

To study the variation of the FeO spectrum with intensity median spectra for different intensities were created (see Sect. 3.1.2). The shape of the quintile spectra is almost identical. They can be transformed into each other by multiplying with a constant factor. Comparing the median spectra to a theoretical spectrum by Gattinger et al. (2011) we found three major deviations: a broader main peak, less structure in the main peak and a possible additional emission peak between 0.615 and 0.626 µm. Reasons for the deviations could be additional airglow emission from NiO and NO+O (Evans et al., 2011). Evans et al. (2011) pointed out, that the ratio of NiO and FeO is highly variable (0.3-2.3). In this study we found the maximum and average possible contamination of NiO to be $\sim 30\%$ and $\sim 2\%$ of the FeO main peak emission, respectively. To explain the broader main peak one could consider collisional excitation and deactivation. However, the radiative lifetime ($\sim 400$ ns) is much smaller than the collisional lifetime ($\sim 200$ µs) (see Gattinger et al., 2011). On the other hand, uncertainties in the molecular parameters as well as uncertainties in the population distribution of the FeO levels by the $Fe+O_3$ reaction may cause further deviations from the measured spectrum. To find the reason for the excess in the measured spectra compared to Gattinger et al. (2011) further simulations of the FeO spectrum are needed.

The diurnal variability of the FeO emission was studied with respect to the four seasons (Sect. 3.2.2). In general, FeO and Na show a very similar behaviour. None of the seasons show a sudden drop of the FeO emission at the end of the night as observed by Saran et al. (2011). This may be due to the better time resolution of Saran et al. (2011) at the end of the night. In general, it has to be noted that the behaviour of the seasons can vary strongly between the different years.

Finally, we studied the seasonal variation of the FeO emission (Sect. 3.3.2). We derived the





annual and semi-annual amplitudes and phases of FeO and compared these results with the analogous parameters derived for Na. We compared our results for sodium with the study of Takahashi et al. (1995), who found a dominant semi-annual amplitude for Na. Possible reasons for this difference in the relative strength of the annual and semi-annual amplitudes could be a change in latitude (e.g. Fukuyama, 1976, 1977; Wiens & Weill, 1973), longitude, a difference in the sampling or the observing period. Other studies (e.g. Fukuyama, 1976, 1977; Wiens & Weill, 1973) showed that the dominance of the semi-annual amplitude decreases with increasing latitude. These results were satisfactorily reproduced by a whole atmosphere chemistry climate model (see Sect. 3.4), which revealed that a $13\pm3\%$ quantum yield for the $Fe+O_3$ reaction best fitted the entire observational data set, although this may be too large by a factor of 2 since there is evidence that the $O_3$ density in the upper mesosphere predicted by WACCM is about 50% too small. Using the reconstructed spectrum (see Sect. 3.1.2) as reference instead of the Gattinger et al. (2011) template we derive a quantum yield of $16\pm6\%$.

*Acknowledgements.* This project made use of the ESO Science Archive Facility. X-shooter spectra from different observing programmes of the period from October 2009 to March 2013 were analysed. This publication is supported by the Austrian Science Fund (FWF). S. Unterguggenberger and S. Noll receive funding from the FWF project P26130. W. Kausch is funded by the project IS538003 (Hochschulraumstrukturmittel) provided by the Austrian Ministry for Research (BMWFW). W. Feng and J. M. C. Plane are funded by the European Research Council (project number 291332 - CODITA).

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



**Table 1.** Coefficients of harmonic analysis

|     | Method | $A_0$ [R] | $A_1$ [R] | $A_2$ [R] | $\varphi_1$ [DOY] | $\varphi_2$ [DOY] |
|-----|--------|-----------|-----------|-----------|-------------------|-------------------|
| FeO | best fit | 22.3 | 3.6 (16) | 7.1 (31) | 180 | 108 |
|     | mean | 23.2 | 4.0 (17) | 6.2 (27) | 177 | 106 |
|     | $\sigma$ | 1.1 | 1.9 | 1.6 | 35 | 8 |
| Na  | best fit | 40.4 | 9.9 (25) | 9.9 (25) | 192 | 108 |
|     | mean | 39.9 | 9.9 (25) | 11.9 (30) | 191 | 108 |
|     | $\sigma$ | 1.1 | 0.5 | 2.4 | 8 | 1 |
|     | Cachoeira Paulista | 33 | 9.2 (28) | 16.3 (41) | 192 | 109 |

$A_0$ denotes the annual mean intensity in Rayleigh.
$A_1$ is the amplitude of the annual oscillation in Rayleigh. In parentheses the relative value of $A_1$ to $A_0$ is given.
$A_2$ is the amplitude of the semi-annual oscillation in Rayleigh. The values in parentheses are the same as for $A_1$.
$\varphi_1$ is the phase shift of the annual oscillation in DOY.
$\varphi_2$ is the phase shift of the semi-annual oscillation in DOY.
The column 'Method' refers to the fit used for the model. Best fit was achieved with a least squares fit. Mean is the value retrieved with a Bayesian approach with $\sigma$ as the standard deviation.
The comparison data are from Cachoeira Paulista (Takahashi et al., 1995).



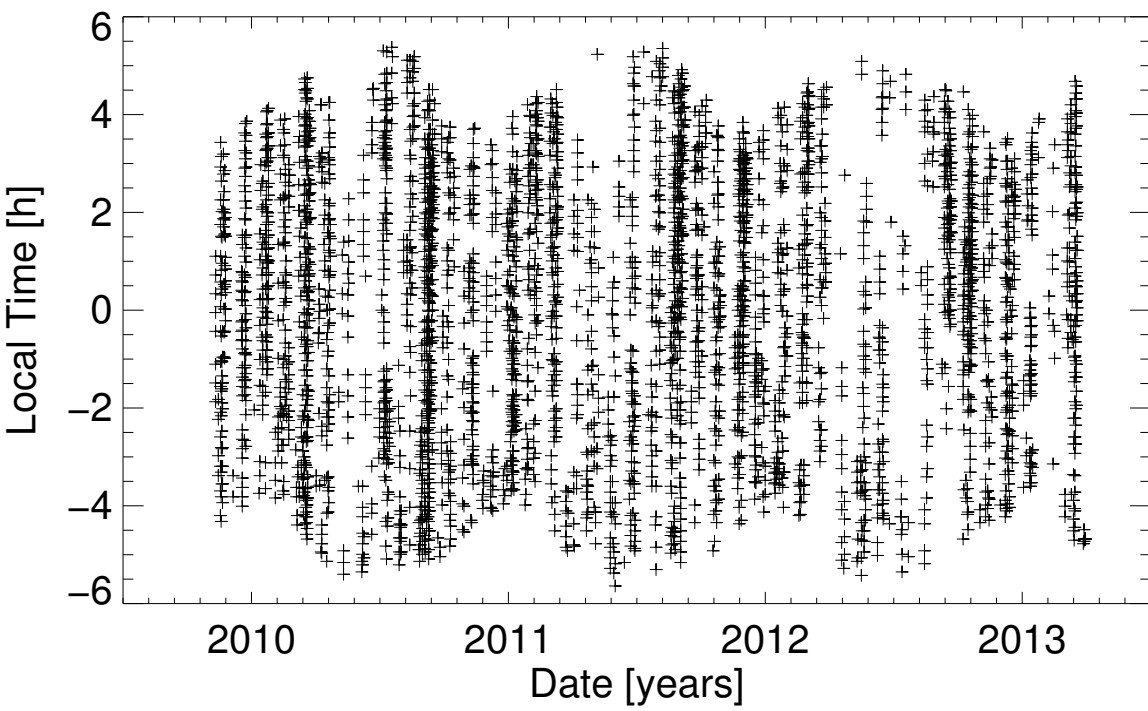

**Figure 1.** Coverage of the X-shooter data set between October 2009 and March 2013. The black crosses show all available X-shooter observations which have an exposure time of at least 10 min. The ordinate shows the observing time with 0 labelling local midnight.





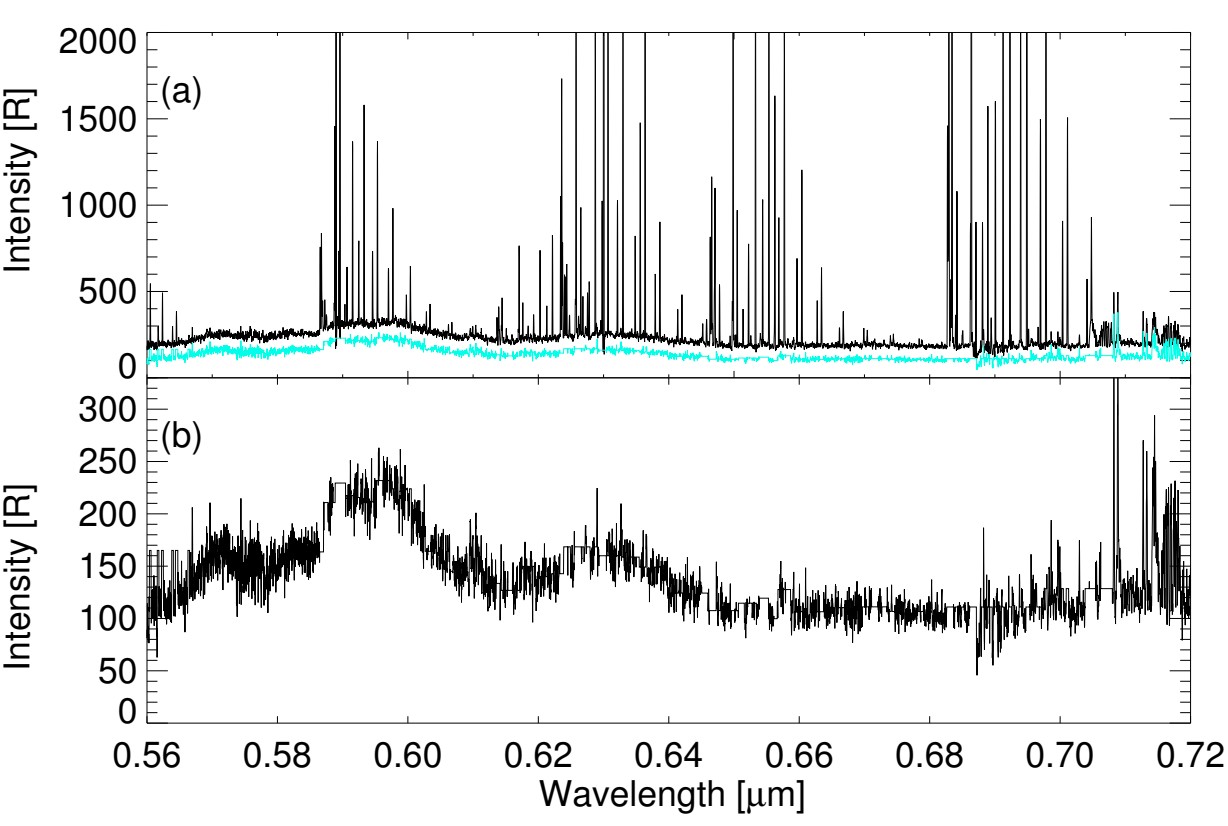

**Figure 2.** Example of a FeO spectrum. Panel (a) displays a raw night-sky spectrum with all its emissions. The FeO continuum, with possible contamination by NiO and NO+O, is overplotted in cyan. Panel (b) shows the same spectrum after removal of all airglow emission lines, zodiacal light, scattered star- and moon light. The spectrum has an exposure time of 3 600 sec and a resolving power of 7 450. The ordinate shows the emission intensity in Rayleighs (1R = $10^6$ photons cm$^{-2}$ s$^{-1}$) while the abscissa shows the FeO emission regime in µm.





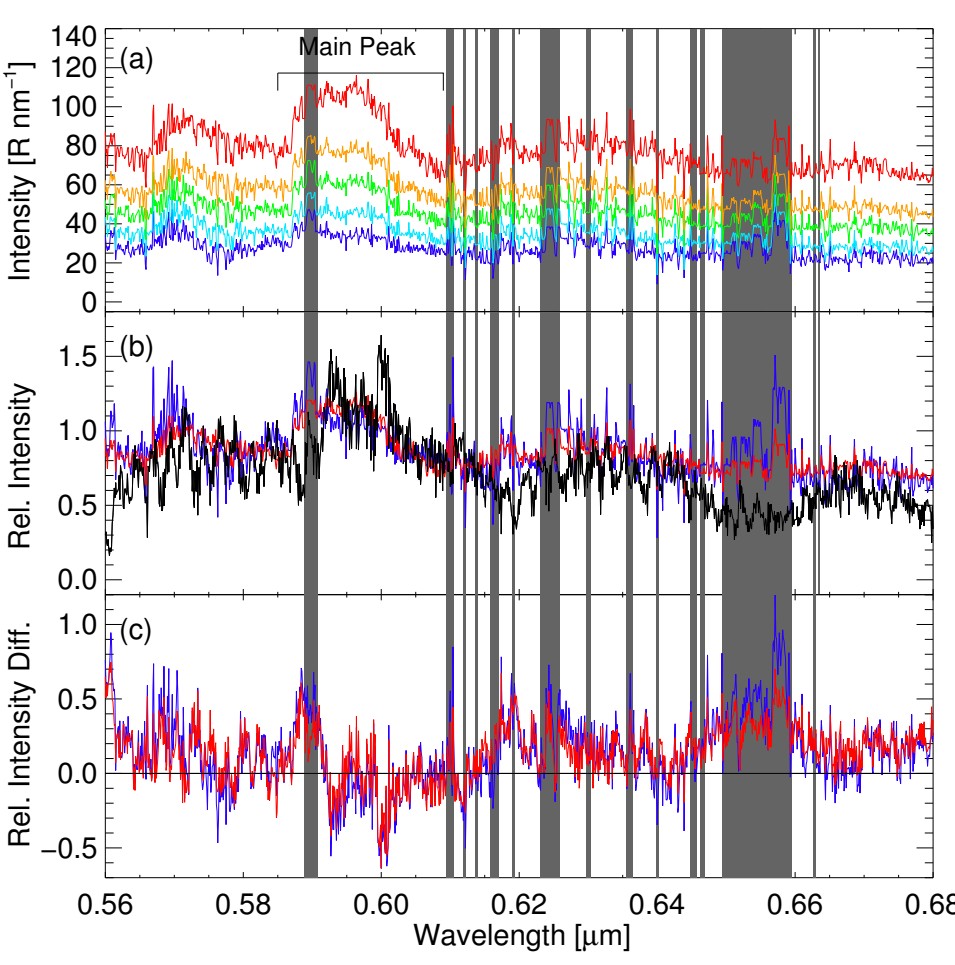

**Figure 3.** Panel (a) displays the five median spectra. Panel (b) illustrates the highest and the lowest quintile median spectra from panel (a) as well as the theoretical spectrum from Gattinger et al. (2011) in black (see Sect. 3.1.1), with the average main peak emission normalised to one. Panel (c) displays the deviation of the two selected median spectra from the theoretical spectrum by subtracting the theoretical spectrum from the median spectra. The colours of the quintile spectra range from blue (median spectrum of the lowest intensity spectra) to red (median spectrum of the highest intensity spectra).

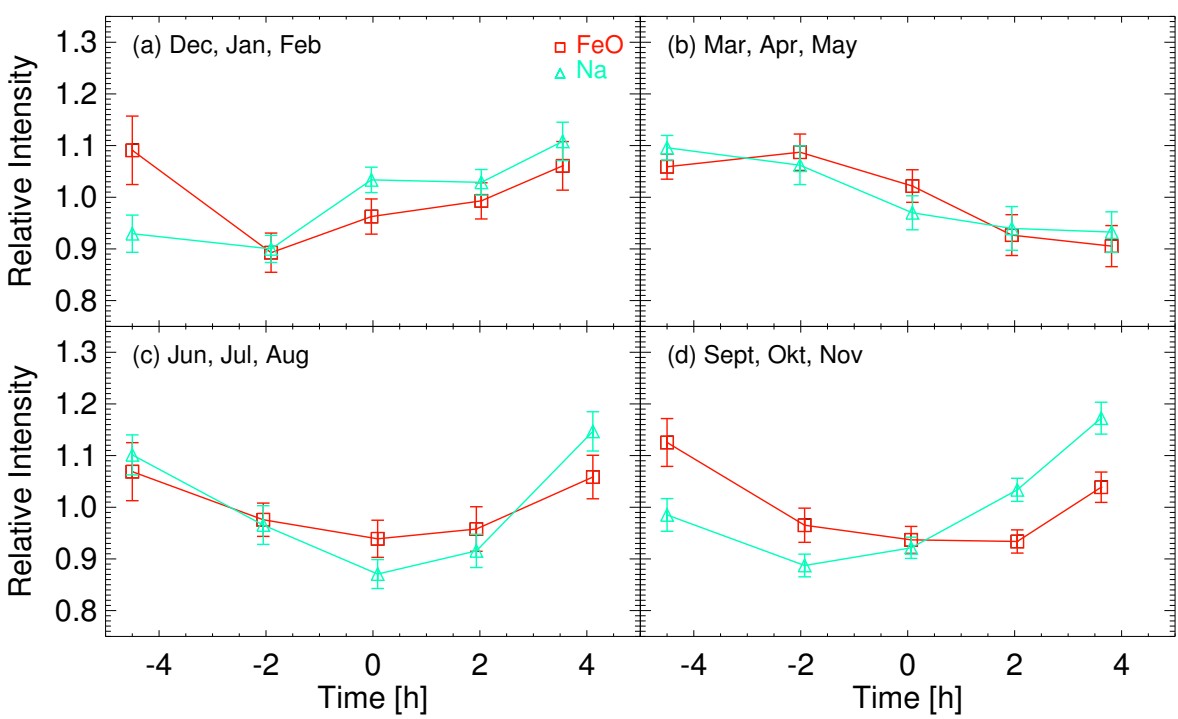

**Figure 4.** Diurnal variability with respect to season. The colours indicate the emitter: FeO (red squares), Na (cyan triangles). The symbols are placed at the mean time of all the measurements in each bin with respect to midnight. The ordinate shows the intensities relative to the nocturnal average.

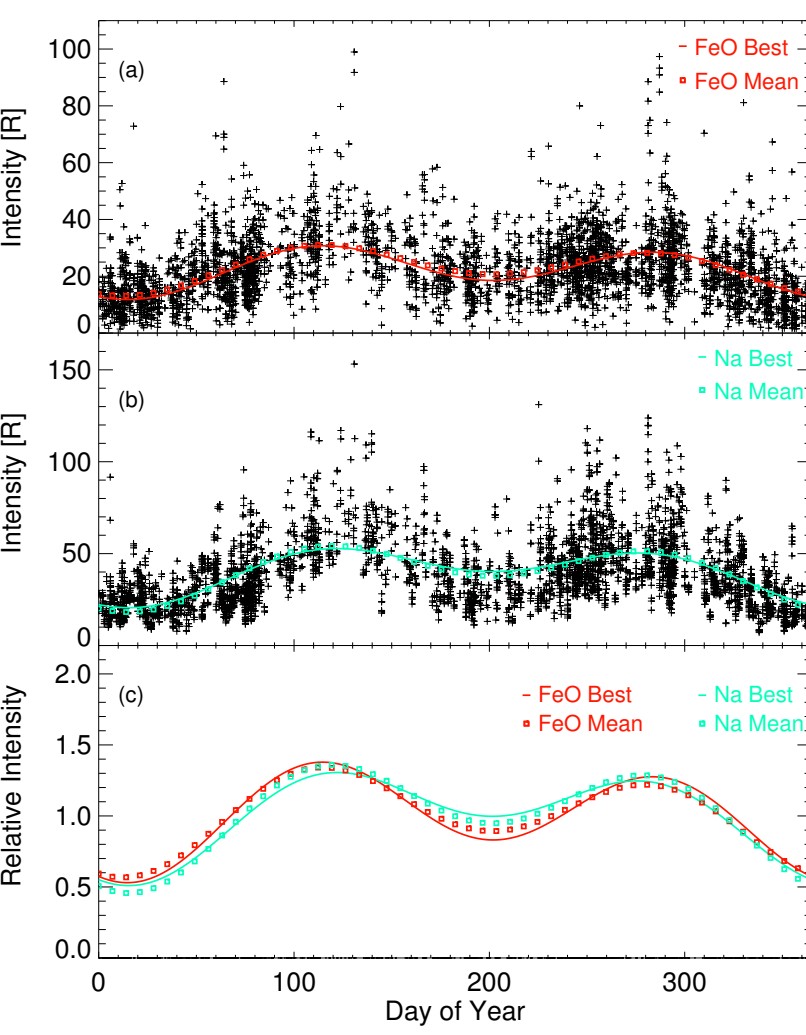

**Figure 5.** Harmonic Analysis: This figure shows the results of the harmonic analysis of FeO in panel a and Na in panel b. The best-fit approach is plotted with a solid line, while the squares indicate the Bayesian (mean) solution. The coefficients of the best fit and mean are given in Table 1. Panel c compares the behaviour of FeO and Na relative to their annual mean. The colours, lines and symbols are the same as in the panels above.



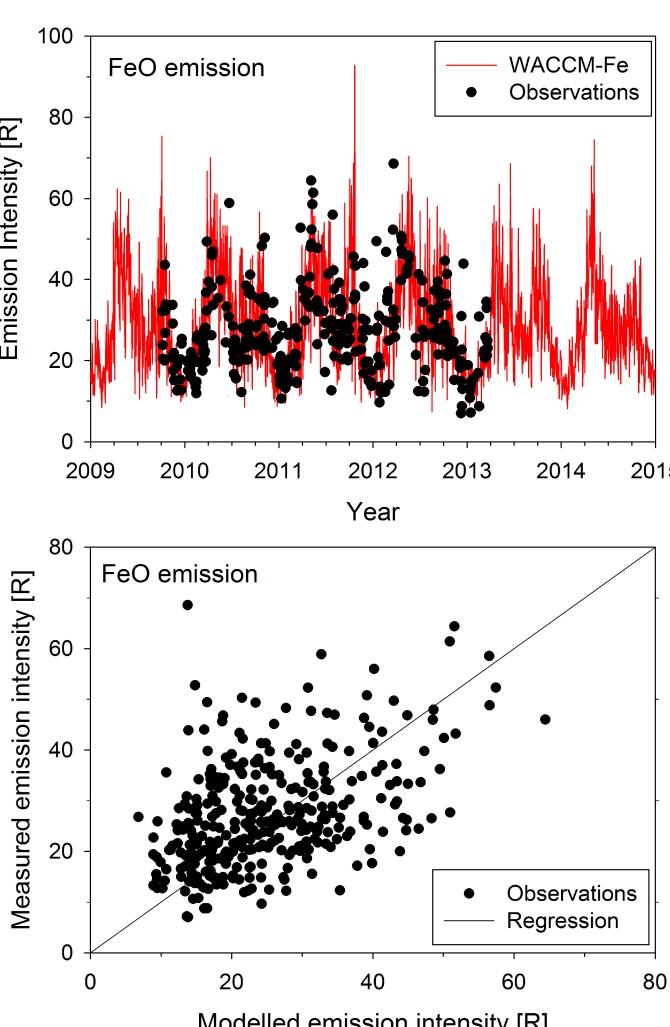

**Figure 6.** Comparison of the measured and modelled FeO emission intensity. Top panel: absolute intensities over the period 2009-2015. Bottom panel: correlation between the measured and modelled intensities. Note that the modelled intensities are scaled to produce a 1:1 regression line through the origin.





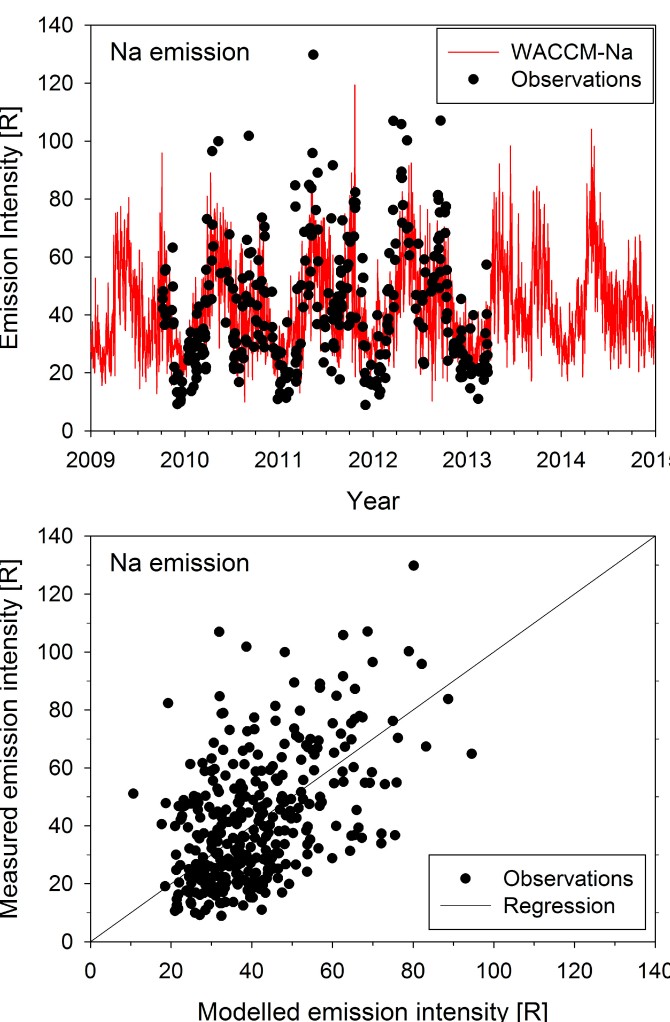

**Figure 7.** Comparison of the measured and modelled Na emission intensity. Top panel: absolute intensities over the period 2009-2015. Bottom panel: correlation between the measured and modelled intensities. Note that the modelled intensities are scaled to produce a 1:1 regression line through the origin.



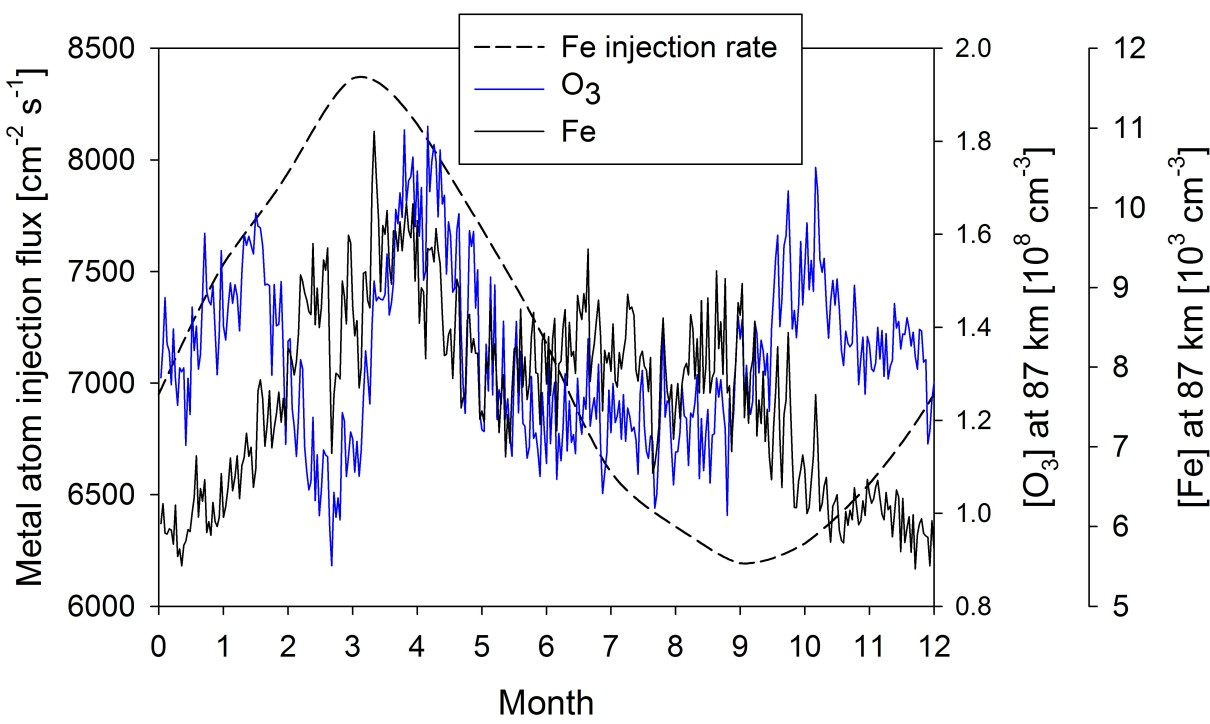

**Figure 8.** Monthly variations of the injection flux of Fe from meteoric ablation, and the concentrations of $O_3$ and Fe at 87 km (averaged from 2009 to 2015) predicted by WACCM.