# Peer review of "Measuring FeO variation using astronomical spectroscopic observations"

_Atmospheric Chemistry and Physics, 2016_

## Referee Comment (RC1) · C. von Savigny (Referee) · 6 Nov 2016

General comments:

This is an interesting manuscript dealing with ground-based observations of two terrestrial nightglow emission features, i.e. the FeO orange bands and the well-known Na D-lines. The spectral observations were carried out with the X-shooter spectrograph at the Very Large Telescope in Chile. The FeO and Na emissions show similar diurnal and seasonal variations. Comparisons with WACCM model simulations allow empirical estimation of the (effective) quantum yields for the two emissions, which are not well known. The paper is of interest to the aeronomy community and is in general very well written. A few paragraphs and sentences are difficult to follow (see specific comments below). I don't have any major objections against the publication of this manuscript

and recommend publication subject to minor revisions. I ask the authors to consider the specific comments listed below.

Specific comments:

Page 3, line 20: "as A pseudo-continuum" ?

Page 7, line 9: "We tested our results with respect to the data distribution over the year by introducing equally spaced bins"

I don't fully understand what you mean here? How many bins were used? How wide/long were they? Do they have to be equally spaced?

Figure 2: I have some questions about this Figure:

(a) Is the ordinate label / unit correct? The plot shows the spectral intensity, so the unit should be R / (wavelength unit), e.g. R/nm, right? This applies to both panels.

(b) You write that the cyan line in the top panel corresponds to the FeO continuum, while the black line shows the raw spectrum. What is the origin of the offset between the two lines? Is it possible that the cyan line is offset by 100 R for better visibility? If yes, this is not mentioned, as far as I can tell.

Page 9, line 19: Evans et al. found a NiO/FeO ratio of 0.05 to 0.3 and Gattinger et al. a ratio of 2.3. Is the large difference between these results understood?

Page 9, line 26/27: "we find a maximum contribution of 31% to the mean peak"

I suggest adding "of the FeO emission" here (this is what you mean, right?)

Page 10, line 2: "Scaling the main peak emission from Gattinger et al. (2011) to the whole spectrum we obtain a value of 3.9%"

I think some pieces of information are missing here. What "value" do you mean? Even after reading the sentence several times, I'm not sure I interpret it correctly. Please clarify.

Page 10, line 10: "where the main peak amounts to 3.3 +-0.8%"

3.3 % of what? This is related to the previous point. Please clarify.

Page 11, line 6: "In general, the Na and FeO emission show similar diurnal variation within their combined errors, i.e. Figs. 4a, c, and d."

This statement is also true for 4b, and even more so than for, e.g. 4d or 4a.

Next sentence: "The intensities of FeO and Na decline at the beginning of the night and rise towards sunrise"

This is not true for 4b. I think you intend to only mean panels a, c d here, right? But this is not explicitly stated by this or the previous sentence (the phrase "i.e. Figs. 4a, c and d" does not imply that).

Page 11, last line: "The best fit approach (chi^2_min) relies on the grid size and does not provide uncertainties"

After reading the entire paragraph I understand what you mean, but there are different "best-fit" approaches. You create arrays with possible fit parameters and then determine chi^2 for each set of possible combinations. One may also use – and I think this is generally done – numerical routines to find the optimum fit values in a least-squares sense. I suggest mentioning at the beginning that you don't use a numerical scheme to minimize chi^2. Otherwise, the reader has difficulties understanding what you mean by "relies on the grid size" – this is not correct for the numerical methods. Also the numerical methods will generally provide uncertainty estimates.

Page 12, line 8: "parmeter" -> "parameter"

Page 12, line 22: "stronger .. amplitude" -> "larger .. amplitude" ?

Page 13, line 8: "at the end of May"

Isn't it rather the end of April?

Page 14, line 23: "by convolving"

Is this really a convolution in the mathematical sense? This may well be the case, but I'm not entirely sure.

Page 15, line 3: "convolving"

Same as above point.

Page 15, line 9: "Fig, 9" -> "Fig. 9"

Page 15, last sentence: I think it's also worth mentioning that Clemesha et al. (1995) performed a minimization of the differences between the observed Na emission rates and model simulations, which resulted in a value of f = 0.093. Also, in our recent manuscript (von Savigny et al., First mesopause Na retrievals from satellite Na D-line nightglow observations, Geophys. Res. Lett., revised, 2016) we find an optimum value of f = 0.09, when comparing Na retrievals from SCIAMACHY Na nightglow observations with independent satellite observations (SCIAMACHY dayglow and GOMOS stellar occultation). I should point out that we varied f in steps of 0.01 to find the optimum value – an approach that can be refined. In any case, I find it encouraging that your results on the value of the effective quantum yield are in good overall agreement with the von Savigny et al. (2016) value and with Clemesha et al. (1995).

Fig. 4, caption, line 1: I Suggest replacing "with respect to season" by "for different seasons". Same line: space in ".The" missing.

Figure 5: The symbols (squares) are hardly visible in the printout. Please increase the symbol size.

Page 16, line 2: "airglow emissionS" ?

Page 16, line 23: "None of the seasons showS"

Page 23, table caption, line 2: "the relative value of A1 .."

Suggest adding "in percent" to read "the relative value (in percent) .."

Reference list: the reference list contains a fair number of typos and inconsistencies. I probably didn't catch all of them. Please check the list again carefully. A general issue: periods are missing at the end of all references. In addition, the spacing between initials is not consistent between the references.

Page 17, line 24: "D.:Chemiluminescent"

Page 17, line 25: "..0 ,2006" -> "..0, 2006."

Page 18, line 1: "J. M. C. Plane, J. M. C."

Page 18, line 14: delete "&"

Page 18, lines 24, 27 and 30: "Phy." -> "Phys."

Page 18, line 30: "variation,J."

Page 19, lines 18 and 21: "Phy." -> "Phys."

Page 20, lines 1 and 2: semicolons used as separators between authors, rather than commas.

Page 20, line 9: "Res- Atmos."

Page 21, line 1: "Plane,J . M."

Page 21, line 10: "J. M. C. Plane, J."

Page 21, line 20: "Mlynzak" -> "Mlynczak"

Page 22, line 9: von Savigny (2012) is not cited in the manuscript, as far as I can tell (But I'm certainly happy if you cite it ..)

Page 22, line 14: "variationsof"

---

## Referee Comment (RC2) · T. P. Viehl (Referee) · 14 Feb 2017

General comments

The paper presents spectroscopic observations of FeO and Na nightglow emissions by the X-shooter instrument at the VLT / Paranal Observatory, Chile. The observations are analysed on diurnal as well as seasonal scales and compared to theoretical considerations. The seasonal variation of the emissions is very satisfactorily reproduced by an atmospheric chemistry model. This analysis reveals new insights about FeO in the MLT and the quantum yields of the relevant emissions.

The paper presents new data and insights which are relevant to the field and well suited for publication in ACP. The methods and assumptions are valid and clearly outlined. In general, the experiment and the calculations are sufficiently described. Some sugges-

tions are given below to improve the description further. The authors give proper credit to related work and clearly indicate their own contribution. The overall presentation is well structured and clear. Some suggestions to improve the presentation further are given below. The title reflects the contents of the paper and the abstract provides a concise and complete summary.

I recommend publication of this interesting manuscript after minor revision and ask the authors to address the following comments and suggestions.

Specific comments and suggestions

Page 2, line 24: Suggestion: change "source of THE metals" to simply "source of metals", "source of metal layers", "source of meteoric metals" or similar, as this is the first account of mentioning "the metals"

Page 3, line 12: It appears slightly odd to me to refer to sodium as "a good CANDI-DATE" since observations of Na are well established and not only theoretically considered. Suggestion: "a good candidate" -> "well established", "commonly used", or similar

Page 5, line 16: Can you provide more information about the criteria of the "additional quality checks" applied to the data? Are these implemented in the pipeline (Modigliani et al., 2010) or are additional reductions performed, e.g., by discarding spectra with obvious distortions through technical problems?

Page 5, lines 19/20: "...an adapted version of pipeline v2.6.8 of the ESO public pipeline..." Suggestion: change to "...an adapted version of the public ESO pipeline v2.6.8..."

Page 5, line 28: I'm not familiar with the term "echelle orders". I suggest changing to "higher diffraction orders of the echelle spectrometer" if this is what is meant. Furthermore, can it be easily explained how the pipeline (i.e., data processing) introduces these as opposed to the instrument? It might be worthwhile adding a further short

explanatory sentence, since this influence does not seem to have been covered in the cited literature.

Page 6, line 12: "FeO is only a faint pseudo continuum component": this doesn't really make sense to me. Suggestion: change to "FeO has only a faint pseudo continuum component" or "the component of FeO to the observed pseudo continuum is only faint" depending on what you want to say here.

Page 6, lines 23&24 and Figure 2: In the text you refer to the exposure time as "roughly 1 hour" and the resolving power as "\approx 7450". In the caption of Figure 2, however, the exposure time is given as precisely "3600 sec" and the resolving power as "7450" (without approx). Please clarify, e.g., by choosing either "roughly 1 hour" or "3600 sec", whichever is correct.

Page 7, line 2: Recommendation: change "this interval" to "the interval" as it is not referred to in the previous sentence.

Page 7, line 14: "which IS according to Gattinger et al. [2011] defined at" -> "which according to Gattinger et al. [2011] IS defined at" ?

Page 7, line 14: "FROM their selected wavelength range" -> "IN their selected wavelength range" ?

Page 7, lines 13/14: Here and throughout the manuscript there are several personal references to studies (i.e., "They found..." instead of "That study found..."). This very much is a stylistic choice of the authors, but I suggest to change those occurrences to the more neutral, impersonal form. In this example, I suggest changing "from their selected wavelength range" to "in the wavelength range of that study" or similar.

Page 7, line 15: I recommend placing "of 6%" between "an error" and "for the FeO main peak"

Page 8, line 24: While the observed FeO spectra indeed match the theoretical work of Gattinger et al. with "good overall agreement", I recommend to add a note that some

parts show a difference in relative intensity of more than 50% (in particular at around 590 nm and 600 nm, well within the main peak).

Page 10, line 2/3: "...we obtain a value of 3.9%" ...of what? Do you mean to say "...we find that FeO contributes 3.9% to the overall spectrum." ?

Page 10, line 27: "...are the low statistic bins." Suggestion: change to "...have the lowest statistic." or "...contribute the fewest data points." or similar.

Page 11, line 6/7: "show A similar" or "show similar ... variationS" ?

Page 11, lines 11-15: This description is not very clear to me. If 2010 shows a different behaviour than 2011&2012, does this mean the effect in those years would be even stronger than in the combined data shown or was 2010 excluded from the plot? Similarly, were the data from September and November 2011 excluded for the reason given or does this imply that the data from these months decreases the effect shown?

Page 14, line 10: Suggestion: "...rates were using the..." -> "...rates were CALCU-LATED / ESTIMATED using the..."

Page 16, line 7: "The shape of the quintile spectra is almost identical." Identical to what or during which periods? I assume it is meant identical to each other.

Figure 3: Please use a lighter shade of grey for regions where the correction might not have been performed accurately. The contrast of the black and blue curves to the grey shaded areas is very low. It is furthermore slightly distracting to have features as prominent as the dark grey areas in the figure without a description in the figure or its caption.

Technical comments

Page 6, line 6: "and as well as": choose either "and" or "as well as"

Page 7, line 28: "are discussed" -> "is discussed" ?

Page 8, line 6: "merge" -> "merged" or "merging"

Page 8, line 15: "normalize" -> "normalized"

Page 8, line 15&16: Throughout the manuscript, you seem to prefer British English over American English. While "normalized" is probably acceptable in BE, you might consider changing to "normalised" for consistency here.

Page 12, lines 4/5: "...between 0 and 0.65 x the maximum intensity" Change to "...between 0 and 0.65 OF the maximum intensity" or consider using percentages as done previously

Page 12, line 20: "...squareS the Bayesian..."

Some comma errors, e.g. Page 16, line 6: "...with intensity, median..." Page 16, line 12/13: "...pointed out that..."

Figure 4(d): "Okt" -> "Oct"

---

## Author Comment (AC1) · 5 Mar 2017

**Response to interactive comment by C. von Savigny**

**General comments:**

*This is an interesting manuscript dealing with ground-based observations of two terrestrial nightglow emission features, i.e. the FeO orange bands and the well-known Na D-lines. The spectral observations were carried out with the X-shooter spectrograph at the Very Large Telescope in Chile. The FeO and Na emissions show similar diurnal and seasonal variations. Comparisons with WACCM model simulations allow empirical estimation of the (effective) quantum yields for the two emissions, which are not well known. The paper is of interest to the aeronomy community and is in general very well written. A few paragraphs and sentences are difficult to follow (see specific comments*

[Figure]

*below).  I don't have any major objections against the publication of this manuscript
and recommend publication subject to minor revisions.  I ask the authors to consider
the specific comments listed below.*

**Specific comments:**

*Page 3, line 20: "as A pseudo-continuum"*
The phrase was changed as proposed.

*Page 7, line 9: "We tested our results with respect to the data distribution over
the year by introducing equally spaced bins"*
*I don't fully understand what you mean here?   How many bins were used?
How wide/long were they? Do they have to be equally spaced?*
The bins were equally spaced over the year with each bin having a width of 2 weeks,
which left us with 26 data points over the year.  For clarification the sentence was
edited as follows: 'We tested our results with respect to the data distribution over the
year by introducing equally spaced bins, spanning a fortnight.'

*Figure 2:  I have some questions about this Figure:  (a) Is the ordinate label /
unit correct?  The plot shows the spectral intensity, so the unit should be R / (wave-
length unit), e.g. R/nm, right? This applies to both panels. (b) You write that the cyan
line in the top panel corresponds to the FeO continuum, while the black line shows the
raw spectrum. What is the origin of the offset between the two lines? Is it possible that
the cyan line is offset by 100 R for better visibility? If yes, this is not mentioned, as far
as I can tell.*
Yes, the label on the ordinate was wrong and has been corrected.
The offset between the black and the cyan spectrum results from the data reduction.
With the help of the sky model we corrected the night-sky spectrum for zodiacal light,
scattered star- and moonlight.  This correction causes the offset between the two

spectra. This information was added to the text.

*Page 9, line 19: Evans et al. found a NiO/FeO ratio of 0.05 to 0.3 and Gat-tinger et al. a ratio of 2.3. Is the large difference between these results understood.*
There was a typo with the second reference. Both measurements (OSIRIS and the space shuttle) were mentioned in Evans et al (2011). This was corrected.
In the paper Evans 11 no answer was given to why they found these big discrepancies in their measurements. This information was added to the text: 'These differences were not discussed in detail in Evans et. al. (2011).'

*Page 9, line 26/27: "we find a maximum contribution of 31% to the mean peak"*
*I suggest adding "of the FeO emission" here (this is what you mean, right?)*
Indeed, in single spectra we could find a possible contribution of the main peak by NiO that amount to almost 1/3 of the total main peak intensity. This information was added to the text as follows: 'Using this peak to scale the laboratory NiO spectrum accordingly we find a maximum contribution to the FeO main peak intensity by NiO of 31%... '

*Page 10, line 2: "Scaling the main peak emission from Gattinger et al. (2011) to the whole spectrum we obtain a value of 3.9%"*
*I think some pieces of information are missing here. What "value" do you mean? Even after reading the sentence several times, I'm not sure I interpret it correctly. Please clarify.*
Due to other airglow contamination like different OH bands, NiO and NO+O it is difficult to measure the FeO spectrum. The most reliable part is the main peak. Since we wanted to see how much of the total FeO intensity is contained within the main peak, we took the the theoretical FeO spectrum from Gattinger et al. (2011) and scaled it to the total FeO flux. Hence, the 3.9 % refer to the contribution of the main peak to the total FeO spectrum. '... FeO pseudo-continuum...' was added for clarification.

*Page 10, line 10: "where the main peak amounts to 3.3 $\pm 0.8\%$" 3.3% of what? This is related to the previous point. Please clarify.*

The Gattinger spectrum spans a wavelength range from 0.5 to 0.72 micron. The main peak is only a small part of the total emission. Hence, if one is interested in the total intensity of the FeO emission, it is necessary to to scale the main peak emission to the total emission of FeO. For a better comparison between the spectrum obtained by Gattinger and our reconstructed spectrum we compared the contribution of the main peak to the total pseudo-contiuum. We add '...of the total FeO emission ranging from 0.50 to 0.72 micron...' as clarification to the text.

*Page 11, line 6: "In general, the Na and FeO emission show similar diurnal variation within their combined errors, i.e. Figs. 4a, c, and d." This statement is also true for 4b, and even more so than for, e.g. 4d or 4a. Next sentence: "The intensities of FeO and Na decline at the beginning of the night and rise towards sunrise"*
*This is not true for 4b. I think you intend to only mean panels a, c d here, right? But this is not explicitly stated by this or the previous sentence (the phrase "i.e. Figs. 4a, c and d" does not imply that).*

Thank you very much for finding this. We corrected for the errors.

*Page 11, last line: "The best fit approach ($\chi 2_{min}$) relies on the grid size and does not provide uncertainties"*
*After reading the entire paragraph I understand what you mean, but there are different "best-fit" approaches. You create arrays with possible fit parameters and then determine chiË Ę 2 for each set of possible combinations. One may also use – and I think this is generally done – numerical routines to find the optimum fit values in a least-squares sense. I suggest mentioning at the beginning that you don't use a numerical scheme to minimize $\chi 2$. Otherwise, the reader has difficulties understanding what you mean by "relies on the grid size" – this is not correct for the numerical methods. Also the*

*numerical methods will generally provide uncertainty estimates.*
A clarification was added to the text as follows:
'Since the best-fit approach ($\chi 2_{min}$) is not done with a numerical scheme but makes use of a parameter grid, it relies on the grid size and does not provide uncertainties.'

*Page 12, line 8: "parmeter" -> "parameter"*
The typo was corrected.

*Page 12, line 22: "stronger .. amplitude" -> "larger .. amplitude" ?*
The word was corrected.

*Page 13, line 8: "at the end of May"*
*Isn't it rather the end of April?* We checked the data again and the maximum is on DOY 116, which corresponds to the end of April. The correction was applied to the text.

*Page 14, line 23: "by convolving" Is this really a convolution in the mathematical sense? This may well be the case, but I'm not entirely sure.*
Convolved was changed into 'by combining in quadrature'.

*Page 15, line 3: "convolving"*
*Same as above point.*
Convolved was changed into 'by combining in quadrature'.

*Page 15, line 9: "Fig, 9" -> "Fig. 9"*
The typo was corrected.

*Page 15, last sentence: I think it's also worth mentioning that Clemesha et al. (1995) performed a minimization of the differences between the observed Na emission rates and model simulations, which resulted in a value of f = 0.093. Also, in our*

*recent manuscript (von Savigny et al., First mesopause Na retrievals from satellite Na D-line nightglow observations, Geophys. Res. Lett., revised, 2016) we find an optimum value of f = 0.09, when comparing Na retrievals from SCIAMACHY Na nightglow observations with independent satellite observations (SCIAMACHY dayglow and GOMOS stellar occultation). I should point out that we varied f in steps of 0.01 to find the optimum value – an approach that can be refined. In any case, I find it encouraging that your results on the value of the effective quantum yield are in good overall agreement with the von Savigny et al. (2016) value and with Clemesha et al. (1995).*
The additional information was added at the end of the result section.

*Fig. 4, caption, line 1: I Suggest replacing "with respect to season" by "for different seasons". Same line: space in ".The" missing.*
The space was added and the phrase changed to the suggested version.

*Figure 5: The symbols (squares) are hardly visible in the printout. Please increase the symbol size.*
The plot was edited.

*Page 16, line 2: "airglow emissionS" ?*
The typo was corrected.

*Page 16, line 23: "None of the seasons showS"*
The typo was corrected.

*Page 23, table caption, line 2: "the relative value of A1 .."*
*Suggest adding "in percent" to read "the relative value (in percent) .."*
Thank you for the suggestion, it was put into the table caption as 'in per cent' (British English).

*Reference list: the reference list contains a fair number of typos and inconsistencies. I probably didn't catch all of them. Please check the list again carefully. A general issue: periods are missing at the end of all references. In addition, the spacing between initials is not consistent between the references.*
We went through the references and corrected for mistakes and inconsistencies.

*Page 22, line 9: von Savigny (2012) is not cited in the manuscript, as far as can tell (But I'm certainly happy if you cite it ..)*
Sorry for that mistake. This citation was from an older version which still included OH measurements. However, your recent letter is now a part of the discussion section.

———————————————

---

## Author Comment (AC2) · 5 Mar 2017

**Response to interactive comment by T. P. Viehl**

**General comments** *The paper presents spectroscopic observations of FeO and Na nightglow emissions by the X-shooter instrument at the VLT / Paranal Observatory, Chile. The observations are analysed on diurnal as well as seasonal scales and compared to theoretical considerations. The seasonal variation of the emissions is very satisfactorily reproduced by an atmospheric chemistry model. This analysis reveals new insights about FeO in the MLT and the quantum yields of the relevant emissions.*
*The paper presents new data and insights which are relevant to the field and well suited for publication in ACP. The methods and assumptions are valid and clearly*
*outlined. In general, the experiment and the calculations are sufficiently described. Some suggestions are given below to improve the description further. The authors give proper credit to related work and clearly indicate their own contribution. The overall presentation is well structured and clear. Some suggestions to improve the presentation further are given below. The title reflects the contents of the paper and the abstract provides a concise and complete summary.*

*I recommend publication of this interesting manuscript after minor revision and ask the authors to address the following comments and suggestions.*

**Specific comments and suggestions**

*Page 2, line 24: Suggestion: change "source of THE metals" to simply "source of metals", "source of metal layers", "source of meteoric metals" or similar, as this is the first account of mentioning "the metals"*

Thanks for the suggestions. The phrase was changed to 'source of metal layers'.

*Page 3, line 12: It appears slightly odd to me to refer to sodium as "a good CANDIDATE" since observations of Na are well established and not only theoretically considered. Suggestion: "a good candidate" -> "well established", "commonly used", or similar*

'A good candidate' was meant with respect to astronomical techniques since the Na doublet can already be detected with small telescopes and low-to-medium resolution spectrographs. The phrase was changed to 'commonly used'.

*Page 5, line 16: Can you provide more information about the criteria of the "additional quality checks" applied to the data? Are these implemented in the pipeline (Modigliani et al., 2010) or are additional reductions performed, e.g., by discarding spectra with obvious distortions through technical problems?*

The spectra were furthermore checked for residuals introduced by the pipeline reduction (residuals related to higher orders, mentioned in the paper), contamination of the continuum by astronomical objects as well as a check on the reliability of the OH(9-4) intensity measurement. Furthermore, the continuum was checked for unreasonable behaviour like steep sudden steps in the spectrum. A paragraph with this information was added to the manuscript.

*Page 5, lines 19/20: "...an adapted version of pipeline v2.6.8 of the ESO public pipeline..." Suggestion: change to "...an adapted version of the public ESO pipeline v2.6.8..."*
The suggested change was implemented.

*Page 5, line 28: I'm not familiar with the term "echelle orders". I suggest changing to "higher diffraction orders of the echelle spectrometer" if this is what is meant. Furthermore, can it be easily explained how the pipeline (i.e., data processing) introduces these as opposed to the instrument? It might be worthwhile adding a further short explanatory sentence, since this influence does not seem to have been covered in the cited literature.*
Echelle order is indeed an astronomical term. At the position of its first occurrence it is now changed to the more detailed '...high diffraction orders of the echelle spectrograph...'
The main issue was the separation of sky and astronomical object. Applying different extraction methods implemented in the pipeline led to different results. X-shooter either traces the light profile of the astronomical source to separate the object from the sky or uses a predefined window to distinguish between sky and object.
We found that the difference between the spectra for the latter and the former method can be used to construct a template for the order-related contamination (periodic wave pattern). This template could then be scaled to the individual contamination patterns and finally be subtracted. Here we only considered the sky spectra obtained with the second approach which is more reliable for line emission.

This information was also added to the manuscript: 'Comparing different extraction routines we found a systematic variation in flux with wavelength that shows minima at the central positions of the different echelle orders and maxima in the overlapping regions. A correction spectrum was determined by comparing differently extracted spectra. This spectrum was then scaled to the observation and subtracted. In addition, we checked for continuum contribution by astronomical objects, the reliability of the intensities of various OH lines and for unreasonable behaviour like sudden steps in the spectrum.'

*Page 6, line 12: "FeO is only a faint pseudo continuum component": this doesn't really make sense to me. Suggestion: change to "FeO has only a faint pseudo continuum component" or "the component of FeO to the observed pseudo continuum is only faint" depending on what you want to say here.*
The phrase was changed to '...FeO contributes to the airglow as a faint pseudo-continuum'.

*Page 6, lines 23&24 and Figure 2: In the text you refer to the exposure time as "roughly 1 hour" and the resolving power as " approx 7450". In the caption of Figure 2, however, the exposure time is given as precisely "3600 sec" and the resolving power as "7450" (without approx). Please clarify, e.g., by choosing either "roughly 1 hour" or "3600 sec", whichever is correct.*
It is indeed exactly an hour of observation time and a resolving power of approx 7450. The values are correct and the text changed accordingly.

*Page 7, line 2: Recommendation: change "this interval" to "the interval" as it is not referred to in the previous sentence.*
The suggested change was applied.

*Page 7, line 14: "which IS according to Gattinger et al. [2011] defined at" ->*

*"which according to Gattinger et al. [2011] IS defined at" ?*
The suggested change was applied.

*Page 7, line 14: "FROM their selected wavelength range" -> "IN their selected wave- length range" ?*
The suggested change was applied.

*Page 7, lines 13/14: Here and throughout the manuscript there are several personal references to studies (i.e., "They found..." instead of "That study found...). This very much is a stylistic choice of the authors, but I suggest to change those occurrences to the more neutral, impersonal form. In this example, I suggest changing "from their selected wavelength range" to "in the wavelength range of that study" or similar.*
Thank you very much for the comment. We left the text as it was in the original manuscript.

*Page 7, line 15: I recommend placing "of 6%" between "an error" and "for the FeO main peak"*
The suggested change was applied.

*Page 8, line 24: While the observed FeO spectra indeed match the theoretical work of Gattinger et al. with "good overall agreement", I recommend to add a note that some parts show a difference in relative intensity of more than 50% (in particular at around 590 nm and 600 nm, well within the main peak).*
With this sentence we referred to the agreement of the five quintile spectra with each other. For clarification we added '... with each other ...'
Also your suggestion on the differences between the spectra was implemented.

*Page 10, line 2/3: "...we obtain a value of 3.9%" ...of what? Do you mean to*

say "...we find that FeO contributes 3.9% to the overall spectrum." ?
Due to other airglow contamination like different OH bands, NiO and NO+O it is difficult
to measure the FeO spectrum. The most reliable part is the main peak. Since we
wanted to see how much of the total FeO intensity is contained within the main peak,
we took the the theoretical FeO spectrum from Gattinger et al. (2011) and scaled it to
the total FeO flux. Hence, the 3.9% refer to the contribution of the main peak to the
total FeO spectrum. '... FeO pseudo-continuum...' was added for clarification.

*Page 10, line 27: "...are the low statistic bins." Suggestion: change to "...have
the lowest statistic." or "...contribute the fewest data points." or similar.*
The text was changed to 'contain the fewest data points'.

*Page 11, line 6/7: "show A similar" or "show similar ... variationS" ?*
The text was changed to '... show similar variations'.

*Page 11, lines 11-15: This description is not very clear to me. If 2010 shows a
different behaviour than 2011&2012, does this mean the effect in those years would
be even stronger than in the combined data shown or was 2010 excluded from the
plot? Similarly, were the data from September and November 2011 excluded for the
reason given or does this imply that the data from these months decreases the effect
shown?*
For Fig. 4 all data points were used. Fig. 4 shows the average diurnal behaviour for
the respective seasons in a time span of 3.5 years.
We also wanted to discuss the behaviour for the different years. However, the
corresponding results are not as reliable as the complete picture since the sample size
is much smaller.

*Page 14, line 10: Suggestion: "...rates were using the..." -> "...rates were CAL-
CULATED / ESTIMATED using the..."*

The text was changed to '...rates were calculated using the . . .'.

*Page 16, line 7: "The shape of the quintile spectra is almost identical." Identical to what or during which periods? I assume it is meant identical to each other.* Indeed, we referred to the similarities of the quintile spectra among each other.
The phrase 'to each other' was added for clarification.

*Figure 3: Please use a lighter shade of grey for regions where the correction might not have been performed accurately. The contrast of the black and blue curves to the grey shaded areas is very low. It is furthermore slightly distracting to have features as prominent as the dark grey areas in the figure without a description in the figure or its caption.*
The shades were changed to a lighter grey.

**Technical comments**

*Page 6, line 6: "and as well as": choose either "and" or "as well as"*
The 'and' was eliminated.

*Page 7, line 28: "are discussed" -> "is discussed" ?*
The phrase was changed to 'is discussed'.

*Page 8, line 6: "merge" -> "merged" or "merging"*
The tense was changed.

*Page 8, line 15: "normalize" -> "normalized"*
The tense was changed.

*Page 8, line 1516: Throughout the manuscript, you seem to prefer British En-*

[Figure]

*glish over American English. While "normalized" is probably acceptable in BE, you might consider changing to "normalised" for consistency here.*
The correction for British English was applied.

*Page 12, lines 4/5: "...between 0 and 0.65 x the maximum intensity" Change to "...be- tween 0 and 0.65 OF the maximum intensity" or consider using percentages as done previously*
The suggestion was included in the manuscript.

*Page 12, line 20: "...squareS the Bayesian..."*
The s was added.

*Some comma errors, e.g. Page 16, line 6: "...with intensity, median..." Page 16, line 12/13: "...pointed out that..."*
The suggestion was included in the manuscript.

*Figure 4(d): "Okt" -> "Oct"* The figure was changed.